# RECURRENT DIFFUSION FOR LARGE-SCALE PARAMETER GENERATION

## ABSTRACT

Parameter generation has struggled to scale up for a long time, significantly limiting its range of applications. In this study, we introduce **R**ecurrent diffusion for large-scale **P**arameter **G**eneration, called **RPG**. We first divide the trained parameters into non-overlapping parts, after which a recurrent model is proposed to learn their relationships. The recurrent model's outputs, as conditions, are then fed into a diffusion model to generate the neural network parameters. Using only a single GPU, recurrent diffusion enables us to generate popular vision and language models such as ConvNeXt-L and LoRA parameters of LLaMA-7B. Meanwhile, across various architectures and tasks, the generated parameters consistently perform comparable results over trained networks. Notably, our approach also shows the potential to generate models for handling unseen tasks. This suggests that recurrent diffusion largely increases the practicality of parameter generation.

## 1 INTRODUCTION

Looking back on the journey of deep learning, the scaling up of neural networks is one of the most important keys to its remarkable success across various tasks (Krizhevsky et al., 2012; He et al., 2016; Meta, 2024). In contrast, neural network parameter generation, from HyperNetworks (Ha et al., 2017) to recent diffusion-based methods (Peebles et al., 2022; Wang et al., 2024; Soro et al., 2024), has struggled to scale up effectively, limiting their practical applications. As illustrated in Fig. 1, the scale gap between vision (or language) models and the generated parameters is at least $10^4$, posing significant challenges for this field.

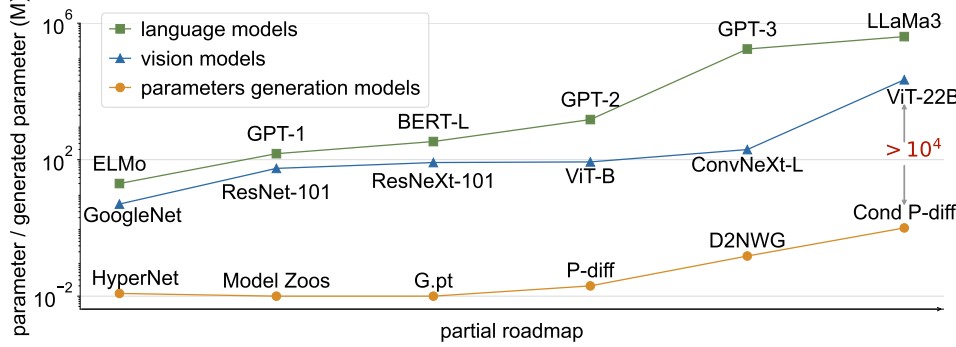

Figure 1: Partial roadmap of vision, language, and parameter generation models. The number of parameters in vision or language models is at least $10^4$ times larger than that of generated parameters.

To figure out the key challenges in scaling up parameter generation, we first analyze its unique requirements. Unlike traditional deep learning models that typically process data such as images or text, parameter generation involves network parameters in the training process. The size of parameters could be significantly larger than images or texts size. This fundamental difference in input format presents a significant challenge when scaling up. As the size of generated parameters increases, the GPU memory requirement quickly becomes prohibitive.

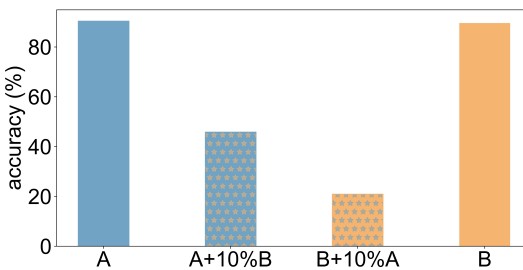

Figure 2: We demonstrate the inherent correlations among neural network parameters by exchanging corresponding partial parameters between two models. Exchanging partial parameters of two models trained with different random seeds leads to significant performance drops.

Recently, p-diff (Wang et al., 2024) and D2NWG (Soro et al., 2024) attempted to address the challenge of balancing memory constraints with large-scale parameter generation. P-diff mainly generates a subset of the entire neural network parameters, while D2NWG employs a two-step process of synthesizing parameter parts and then combining them to form a complete model. However, these methods may overlook the inherent correlations among the parameter parts. To study the impact of these correlations, we conduct an experiment: exchanging partial parameters between two models trained on the same dataset with identical architecture. As illustrated in Fig. 2, the significant performance degradation highlights the critical importance of parameter correlations.

How can we model parameter relationships and leverage them for efficient parameter generation? ViTs often outperform CNNs in vision tasks by modeling patch relationships and capturing global information via self-attention. In language tasks, LLMs use next token prediction to model token relationships, capturing long-range dependencies. Inspired by these approaches, we can consider treating parameter parts as tokens in neural network parameter generation, potentially enabling methods to model inter-parameter relationships and capture dependencies.

Based on the above analysis, we propose **R**ecurrent diffusion for large-scale neural network **P**arameters **G**eneration (**RPG**). Our approach first divides the trained network parameters into a set of non-overlapping parameter parts that are used for supervision (simply called 'tokens' in the following). Subsequently, we use a recurrent model to learn the relationships among the tokens. Finally, the outputs of the recurrent model, as conditions, are fed into a diffusion process to generate the neural network parameters.

Our approach has the following properties: i) With a single GPU, our approach successfully synthesizes large-scale vision models such as ResNet (He et al., 2016), ViT (Dosovitskiy et al., 2021), and ConvNeXt (Liu et al., 2022) series, as well as LoRA (Hu et al., 2021) parameters of LLaMA-7B (Touvron et al., 2023). ii) Across various architectures and tasks, the generated parameters maintain comparable performance to the original models. iii) Empirical evidence suggests our approach has potential for generating models in unseen tasks. We anticipate that this work will inspire future research in large-scale parameter generation.

## 2 HOW TO GENERATE LARGE-SCALE NEURAL NETWORK PARAMETERS?

### 2.1 OVERVIEW

Our approach comprises two key components: parameter tokenization and recurrent diffusion. We show the inference process of recurrent diffusion in Fig. 3. The permutation state and position embedding are fed into the recurrent model. Then, the outputs of the recurrent model serve as conditions for the diffusion process, which generates the entire neural network parameters.

### 2.2 PARAMETER TOKENIZATION

Inspired by the success of language and vision models (Vaswani, 2017; Dosovitskiy, 2020), we propose parameter tokenization that divides network parameters into non-overlapping tokens. Considering the distribution shifts across different layers, we first categorize the trained parameters according to their respective layer indices. Then, we apply normalization (subtracting the mean and dividing by the standard deviation) on each layer. These operations can be formulated as follows,

$$W \xrightarrow{\text{divide by layer}} [w[1], \cdots, w[i], \cdots, w[I]] \xrightarrow[-\mu \text{ and } /\sigma]{\text{normalize}} [\hat{w}[1], \cdots, \hat{w}[i], \cdots, \hat{w}[I]], \quad (1)$$

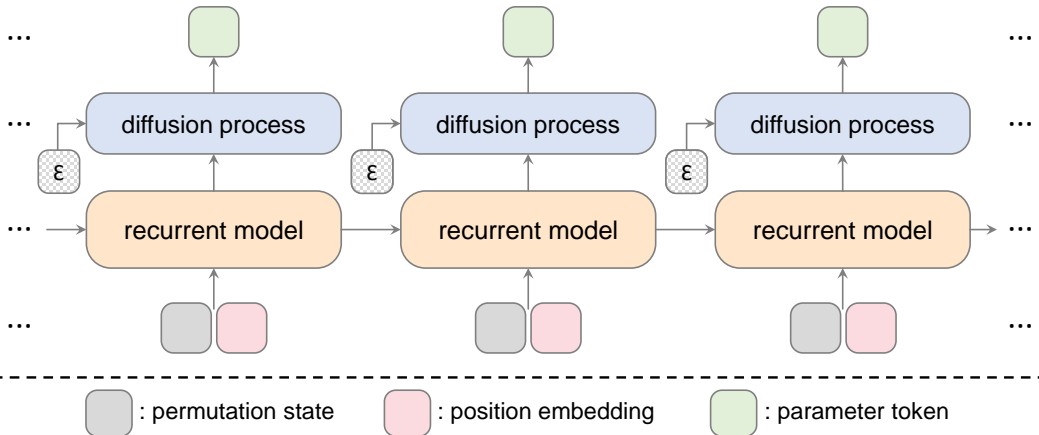

Figure 3: Illustration of the inference of the recurrent diffusion. The recurrent model takes permutation state and position embedding as inputs. The outputs of the recurrent model are then fed into the diffusion process as condition to synthesize the neural network parameters.

where $W$ denotes the trained parameters. $\mu$ and $\sigma$ denote the mean and standard deviation values of parameters. $w_i$ and $\hat{w}_i$ are the original and normalized parameters of the $i$-th layer, respectively.

The number of parameters varies across these layers, which is not conducive to efficient batch training. To this end, we slice each layer parameter into a set of tokens with the same token size, which can be written as follows,

$$\hat{w}[i] \xrightarrow{\text{tokenization}} K[i] = [k_i^1, \cdots, k_i^j, \cdots, \text{padding}(k_i^J)], \tag{2}$$

where the $k_i^j$ represents the $j$-th token of $i$-th layer. For the last token of each layer, we apply padding operation to ensure that all layers have tokens of uniform length. It is worth noting that the padded regions are excluded from the loss calculation.

## 2.3 RECURRENT DIFFUSION

**Permutation state.** Neural network symmetries (Badrinarayanan et al., 2015; Kunin et al., 2021) do not affect the model outcomes but increase the difficulty of learning the parameter distribution. To address this, we introduce a unique state for each trained model $W$ via one-hot embedding. This operation provides a guide for the generated parameters to mitigate the influence of parameter symmetries. For simplicity, we use $S$ to represent the permutation state of $W$.

**Position embedding.** Inspired by ViT (Dosovitskiy et al., 2021), we also encode the layer and token information described in the parameter tokenization (Sec. 2.2) using a two-dimensional sinusoidal position encoding. Specifically, the first dimension encodes the layer index of the token in the original model, while the second dimension encodes the position of the token within its layer. For $i$-th layer parameter tokens $K[i]$, the position embedding can be formulated as follows,

$$K[i] \xrightarrow{\text{position embedding}} e[i] = [e_i^1, \cdots, e_i^j, \cdots, e_i^J], \tag{3}$$

where $e_i^j$ denotes the positional embedding of the parameters belong to $j$-th token of $i$-th layer.

**Recurrent model.** After obtaining the parameter tokens, permutation states, and position embeddings, we use a recurrent model to learn the representation of the parameter tokens. For clarity, we will refer to the output of the recurrent model as the 'prototype' in the following. This operation can be written as follows:

$$P_i^j, H_i^j = f(H_i^{j-1}, e_i^j, S), i \in [1, I], j \in [1, J], \tag{4}$$

where $P_i^j$ and $H_i^j$ denote the prototype and hidden state of the parameters belonging to $j$-th token of $i$-th layer, respectively. $f(\cdot, \cdot, \cdot)$ denotes the state transition function. The structure of the recurrent model is simple. Considering efficiency, we default to using Mamba (Gu & Dao, 2024) followed by an MLP to project the feature dimension to the required size for the diffusion model. We also conduct ablation studies with other recurrent model architectures, such as LSTM (Hochreiter et al., 1997) and transformer with its decoder in a causal manner (Vaswani, 2017).

**Parameter diffusion.** Inspired by p-diff (Wang et al., 2024) and MAR (Li et al., 2024a), we use 1D convolution to build the diffusion model. In this part, the parameter prototypes, serving as conditions, are fed into the diffusion process along with random noise. We optimize our approach through the following equation:

$$L_{\text{diff}} = \mathbb{E}_{t,K,\epsilon}[||\epsilon - \epsilon_\theta(K_t, t, P)||^2], \tag{5}$$

where $K$, $P$, $L_{\text{diff}}$, and $t$ denote parameter tokens, prototypes, diffusion loss, and time step, respectively. $\epsilon$ is the added Gaussian noise and $\epsilon_\theta$ is the denoising network parameterized by $\theta$. Note that, the gradient propagates through $P$ to the recurrent model, implicitly optimizing it as well.

## 3 EXPERIMENTS

In this section, we first introduce our experimental setup for reproducing. Then, we report the results on classification, semantic segmentation, object detection, and commonsense reasoning tasks, respectively. After that, the ablation studies are presented for a better understanding of the benefits of our approach. Finally, we compare our approach with the previous works.

### 3.1 SETUP

**Datasets and architectures.** We mainly evaluate our method across a wide range of tasks, including ImageNet-1K (Deng et al., 2009) for the classification, ADE20K (Zhou et al., 2017) for the semantic segmentation, COCO (Lin et al., 2014) for the object detection, and BoolQ (Clark et al., 2019), PIQA (Bisk et al., 2020), SIQA (Sap et al., 2019), HellaSwag (Zellers et al., 2019), and ARC (Clark et al., 2018) for the commonsense reasoning tasks. To verify the scalability of our approach, we conduct experiments on various architectures with parameter counts ranging from several to hundred million. Details of parameter counts can be found in Tab. 1, 2, 3.

**Trained parameters collection.** We take parameters collection on the ImageNet-1K as an example. To save the cost, we finetune the full parameters of the models released in timm [1] and save 50 checkpoints as the training data. For each checkpoint, we assign a unique permutation state to guide the generated parameters.

**Training details.** We default to using Mamba (Gu & Dao, 2024) as the architecture of the recurrent model. The length of parameter tokens, permutation states, position embeddings, and prototypes is set to 8192. It is worth noting that the permutation states and position embeddings are fixed during the training by default. We also study the influence of the token length, varying it from 1024 to 16384. The parameter diffusion consists of 1D convolutional layers. More details about the model architectures, hyperparameters, and training process can be found in Appendix A.1.

**Inference details.** We input permutation states and position embeddings into the recurrent model to generate the prototypes. Then, the diffusion model utilizes the prototypes as conditions, along with random noises, to synthesize the entire network parameters. We repeat the above process 10 times and report the best, average, medium, minimum, and standard deviation results.

### 3.2 RESULTS OF LARGE-SCALE PARAMETER GENERATION

In this section, we present the results of our approach across a range of tasks including classification, semantic segmentation, object detection&instance segmentation, and language tasks. As most previous works encounter the out-of-memory issue at million-scale parameter generation, we mainly compare with the results from the trained networks, which we denote as 'original'.

---

[1]https://github.com/huggingface/pytorch-image-models

Table 1: We compare with the results of original models across seven architectures on the ImageNet-1K. Our approach successfully generates the entire model parameters that perform comparable results with the original models. **Bold entries** are best results.

| architecture | ResNet-18 | ResNet-50 | ViT-Tiny | ViT-Small | ViT-Base | ConvNeXt-A | ConvNeXt-L |
|---|---|---|---|---|---|---|---|
| params. (M) | 11.7 | 25.6 | 5.7 | 22.1 | 86.6 | 3.7 | 197.8 |
| original acc. (%) | 70.0 | 79.8 | 74.9 | 81.4 | 84.4 | 75.2 | 85.8 |
| best acc. (%) | 69.9 | 79.6 | 75.4 | 80.6 | 84.6 | 74.6 | 85.8 |
| average acc. (%) | 69.5 | 79.5 | 75.3 | 80.5 | 84.4 | 74.4 | 85.5 |
| minimum acc. (%) | 69.0 | 79.4 | 75.2 | 80.1 | 84.2 | 74.2 | 85.2 |
| medium acc. (%) | 69.5 | 79.5 | 75.3 | 80.5 | 84.4 | 74.4 | 85.5 |
| standard deviation | 0.2 | 0.1 | 0.1 | 0.1 | 0.1 | 0.1 | 0.2 |

**Results on ImageNet-1K.** Tab. 1 presents a comparison between our generated models and their 'original' counterparts across seven architectures on ImageNet-1K (Deng et al., 2009). These architectures encompass the ResNet (He et al., 2016), ViT (Dosovitskiy et al., 2021), and ConvNeXt (Liu et al., 2022) series, with parameter counts ranging from 3 to 197 million.

Based on the results in Tab. 1, several crucial observations can be made as follows: i) Our approach successfully generates model parameters at hundred-million scales, overcoming the out-of-memory issues faced by previous works (Peebles et al., 2022; Wang et al., 2024; Soro et al., 2024; Jin et al., 2024). ii) The performances of the generated models are comparable with the original ones. iii) Moreover, our approach exhibits good performance stability as reflected in the small standard deviation. This demonstrates the effectiveness of our approach in hierarchically modeling the parameter relationships.

**Results on ADE20K and COCO.** In addition to the classification task, we also investigate the generalization of our approach to semantic segmentation as well as object detection and instance segmentation tasks. We choose ADE20K (Zhou et al., 2017) and COCO (Lin

Table 2: Accuracy comparison of original and generated parameters on ADE20K and COCO. In these experiments, all models are built based on ViT-Base (Dosovitskiy, 2020).

| method | ADE20K (176.5M params.) | | COCO (110.9M params.) | |
|---|---|---|---|---|
| | mIoU(%) | mAcc(%) | mAP Bbox (%) | mAP Seg (%) |
| original | 47.6 | 58.3 | 43.6 | 39.0 |
| ours | 47.1 | 57.5 | 44.5 | 39.6 |

et al., 2014) as our benchmark datasets. For semantic segmentation, following Zhao et al. (2024), we adopt UperNet (Xiao et al., 2018) as the segmentation model and train it on ADE20K to prepare checkpoints. For object detection and instance segmentation, we finetune ViTDet (Li et al., 2022) on COCO to collect checkpoints and report the results of mAP Bbox and mAP Seg, respectively. All experiments here are conducted based on ViT-B (Dosovitskiy, 2020). Tab. 2 presents the strong generalization of our approach to these two tasks. Specifically, compared to the original models, we achieve comparable or even slightly better results over all the above metrics.

**Results on commonsense reasoning.** To further evaluate the generalization of our approach, we conduct experiments on language tasks. We employ DoRA (Liu et al., 2024), an upgraded version of LoRA (Hu et al., 2022), to fine-tune LLaMA-7B (Touvron et al., 2023) for commonsense reasoning tasks and save the checkpoints as the training data. We report the result comparisons across 7 sub-tasks with rank = 4 in Tab. 3. The generated models consistently yield results comparable to those of the original ones.

Table 3: Accuracy comparison of original and generated DoRA with varying ranks for LLaMA-7B on the commonsense reasoning tasks.

| method | params. (M) | results (%) | BoolQ | PIQA | SIQA | HellaSwag | ARC-e | ARC-c | OBQA |
|---|---|---|---|---|---|---|---|---|---|
| DoRA | 7.8 | original | 64.3 | 71.3 | 66.0 | 53.7 | 64.4 | 49.5 | 63.1 |
| | | ours | 63.1 | 72.0 | 67.5 | 56.7 | 65.3 | 49.7 | 66.0 |

## 3.3 ABLATION STUDIES

In this section, we examine the influences of key factors on our methodology. We primarily present the results of the generated ViT-Tiny (Dosovitskiy et al., 2021) on the ImageNet-1K (Deng et al., 2009), unless stated otherwise.

Table 4: **Ablation experiments of the recurrent model and position embeddings** with ViT-Tiny on ImageNet-1K. Defaults are marked in gray . **Bold entries** are best results.

(a) Recurrent model can largely improve the performance and stability of our method.

|  | best | average | medium |
|---|---|---|---|
| original baseline | 75.2 | 74.9 | 74.9 |
| − recurrent model | fail | fail | fail |
| + recurrent model | 75.4 | 75.3 | 75.3 |

(b) Learnable embeddings performs better, but saving *too many* embeddings is impractical.

| position embedding | best | average | medium |
|---|---|---|---|
| learnable | 75.5 | 75.4 | 75.4 |
| encode by index | 75.4 | 75.3 | 75.3 |
| encode by layer | 75.4 | 75.3 | 75.3 |

**The effect of recurrent model.** We employ the recurrent model to learn the relationship among parameter tokens. To keep other factors consistent, we simply remove the state transition function from the recurrent model for comparison, denoted as '− recurrent model'. The experimental results from Tab. 4(a) confirm that the recurrent model plays a crucial role in parameter generation. Without the state transition function, our approach learns each parameter token individually, overlooking the relationships among these tokens. As a result, the generated parameters perform extremely poorly.

**The manner of position embeddings.** In ViT (Dosovitskiy et al., 2021), the position embeddings are learnable by default. Here, we mainly conduct the experiments with three different position embedding manners and show the details as follows:

• learnable: Initializing with 2D sinusoidal positional encoding and set to be learnable.

• encoded by index: Using 1D sinusoidal positional encoding, irrespective of the original network structure, with indices assigned from front to back.

• encoded by layer (default): Using 2D sinusoidal positional encoding to represent the indices of layer and token, respectively.

As shown in Tab. 4(b), the learnable embeddings perform slightly better than the other two manners. However, we still recommend using fixed position embeddings, as they offer comparable performance while significantly reducing storage requirements compared to the learnable position embedding scheme.

**The manner of tokenization.** Considering the differences among various layers, we divide the parameters into tokens within each layer. Previous works (Wang et al., 2024; Schürholt et al., 2024) employ different processing strategies for parameters. P-diff (Wang et al., 2024) directly flattens all the parameters into a one-dimensional vector, while SANE (Schürholt et al., 2024) divides the parameters by channel within each layer.

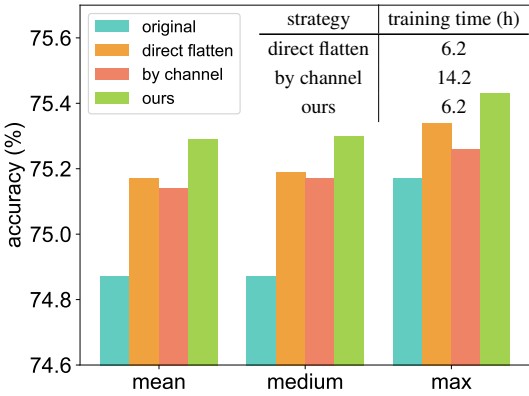

Figure 4: Result and training time comparisons among different tokenization strategies.

We conduct experiments to analyze the three strategies mentioned above and compare their results in Fig. 4. Our default strategy achieves better results than the others. Directly flattening results in a single token containing parameters from different layers, which poses challenges for optimization. Tokenizing by the channel number may result in excessive padding values for each token, as the number of channels is usually much smaller than the default token size.

**The structure of recurrent model.** We mainly explore three structures of the recurrent model, including LSTM (Hochreiter et al., 1997), Transformer (Vaswani, 2017), and Mamba (Gu & Dao,

2024). In Tab. 5, we report the performances, training time, and memory cost of generating ViT-Tiny parameters on ImageNet-1K. All results are obtained on a NVIDIA H100 80G GPU. All three structures can achieve good results. However, considering the training time and memory cost, our default Mamba is the best structure of the recurrent model.

Table 5: We study the characteristics of three recurrent structures. Defaults are marked in gray. **Bold entries** are best results.

| structure \ results (%) | best | average | medium | traning time (hour)↓ | memory (GB)↓ |
|---|---|---|---|---|---|
| LSTM (Hochreiter et al., 1997) | 75.5 | 75.2 | 75.3 | 16.1 | 38.1 |
| Transformer (Vaswani, 2017) | 75.0 | 74.8 | 74.8 | 4.2 | 29.1 |
| Mamba (Gu & Dao, 2023) | 75.4 | 75.3 | 75.3 | 4.1 | 27.8 |

**Token size.** In Tab. 6, we show the results of generating ViT series with different token sizes from 1024 to 16384. The performances of generated models become better as the token size increases. Too small tokens contain limited information that is hard to learn. However, large token sizes lead to substantial memory costs (see in Appendix Fig. 7).

**Efficiency of generating large-scale parameters.** Rapid synthesis of large-scale parameters is crucial for evaluating the practicality of our approach. As illustrated in Tab. 7, we present the time cost for generating models of ViT-Base and ConvNeXt-L across various DDIM (Song et al., 2020) sampling steps. All results are obtained with a single NVIDIA H100 80G GPU. Our approach shows the

Table 6: Accuracy of generated models with the different toke sizes. Large token size performs better on large models.

| model | params. (M) | token size | | | | |
|---|---|---|---|---|---|---|
| | | 1024 | 2048 | 4096 | 8192 | 16384 |
| ViT-Tiny | 5.7 | 0.3 | 70.8 | 75.2 | 75.3 | 69.3 |
| ViT-Small | 22.1 | 0.1 | 0.7 | 80.5 | 80.5 | 80.4 |
| ViT-Base | 86.6 | 0.1 | 0.1 | 0.2 | 45.3 | 84.4 |

capability to generate models within minutes. Notably, even for ConvNeXt-L (197.7 M parameters), we can synthesize the entire parameter within 1.3 minutes. Even with only 5 sampling steps, we can achieve promising results. Meanwhile, the inference memory requirement is approximately 20GB, so our approach can be deployed on NVIDIA GeForce RTX 3090 or similar-level GPUs.

Table 7: GPU memory and inference time comparisons among different diffusion steps. Our approach can generate the entire ConvNeXt-L parameters (197.8 M) in minutes.

| model / memory cost | ViT-Base / 20.7GB | | | | | ConvNeXt-L / 21.6GB | | | | |
|---|---|---|---|---|---|---|---|---|---|---|
| diffusion steps | 5 | 20 | 60 | 100 | 200 | 5 | 20 | 60 | 100 | 200 |
| time (minute) | 0.5 | 0.6 | 0.8 | 1.1 | 1.8 | 0.5 | 0.7 | 1.3 | 2.0 | 3.5 |
| accuracy (%) | 81.1 | 83.3 | 84.4 | 84.4 | 84.3 | 82.1 | 85.0 | 85.5 | 85.3 | 85.3 |

## 3.4 COMPARISONS WITH PREVIOUS METHODS

We compare our approach with four previous works, *i.e.*, $S_{KDE30}$ (Schürholt et al., 2022a), p-diff (Wang et al., 2024), D2NWG (Soro et al., 2024), and SANE (Schürholt et al., 2024). As shown in Tab. 8, our approach consistently achieves the best results on various architectures, while previous works are hard to achieve comparable performances as original models. Another key issue is that the previous works usually fail to generate large-scale neural network parameters.

## 4 CAN WE GENERATE MODEL PARAMETERS IN UNSEEN TASKS?

Until now, experimental results have demonstrated that our approach can efficiently generate large-scale neural network parameters if these models are included in the training set. In this section, we mainly investigate whether our approach has the ability to generate models to tackle unseen tasks.

Table 8: Our approach can generate models with better accuracy, and it can generate a significantly larger number of parameters compared to previous methods. Underline and OOM denote the reproduced results and out-of-memory issue, respectively. The detailed structures of CNN (s) and CNN (m) can be found in Model Zoos (Schürholt et al., 2022)

| method \ accuracy (%) | CNN (s) | CNN (m) | MobileNetV3 | ResNet-18 | ViT-Base |
|---|---|---|---|---|---|
| params. (M) | 0.003 | 0.011 | 4.2 | 11.7 | 86.6 |
| original | 49.0 | 62.1 | 95.4 | 93.9 | 98.7 |
| $S_{KDE30}$ (Schürholt et al., 2022a) | 26.9 | – | OOM | OOM | OOM |
| p-diff (Wang et al., 2024) | 48.8 | 61.9 | OOM | OOM | OOM |
| D2NWG (Soro et al., 2024) | 38.2 | – | 82.2 | – | OOM |
| SANE (Schürholt et al., 2024) | – | 57.9 | – | 68.6 | OOM |
| ours | **49.0** | **62.0** | **89.3** | **93.6** | **98.9** |

## 4.1 EXPERIMENT DESIGNS.

**Build seen and unseen tasks.** To assess our approach's capability in generating models for unseen tasks, we construct various tasks by assigning novel labels to CIFAR-10 categories. Considering the cost of training data collection, we collect multiple binary classifiers on CIFAR-10. Specifically, as illustrated in Fig. 5, we first randomly assign a binary (0 or 1) label for each category of CIFAR-10, respectively. Then, we can obtain $1 \times 10$ binary embedding. The total number of unique binary embeddings should be $2^{10}$, but we need to remove pure 0 and 1 codes. Therefore, there are $2^{10} - 2 = 1022$ valid binary embeddings. Finally, we divide these embeddings into two non-overlapping parts: seen and unseen binary embeddings.

**Collection of the checkpoints.** We use ViT-Tiny to train 1022 binary classifiers on CIFAR-10 with different binary embeddings and save three models for each classifier. These binary embeddings serve as conditioning inputs for the subsequent RPG training process.

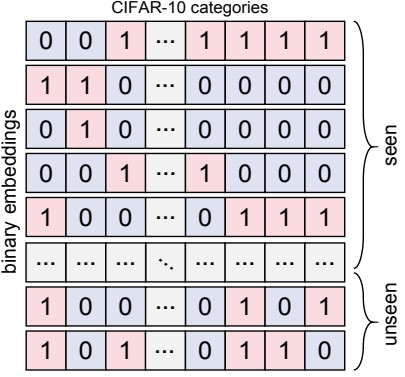

Figure 5: Illustration of building binary embeddings. 0 and 1 denote the labels.

Table 9: Result comparisons between original and generated models on unseen embeddings.

| unseen binary embeddings | original | ours |
|---|---|---|
| 0 1 0 0 0 1 0 1 1 1 | 97.3 | 94.4 |
| 0 1 1 1 1 1 0 1 1 0 | 98.1 | 96.6 |
| 0 0 1 1 1 0 1 1 1 0 | 97.4 | 95.0 |
| 0 1 0 1 1 1 1 1 1 1 | 98.4 | 96.1 |
| 0 0 1 0 0 0 0 0 0 0 | 98.9 | 96.6 |
| 0 0 0 1 1 0 0 1 0 1 | 96.7 | 92.9 |
| 1 1 1 1 1 0 1 0 0 1 | 97.6 | 94.8 |
| 1 0 1 0 0 0 0 0 1 1 | 98.1 | 95.7 |
| 0 1 0 0 0 1 0 1 1 0 | 97.1 | 93.6 |
| 1 1 0 0 0 1 1 0 0 1 | 97.0 | 94.0 |

**Training of RPG.** We only use the checkpoints (training data) trained by the seen binary embeddings as the supervision of RPG. Meanwhile, these embeddings are also fed into RPG as conditional inputs of the recurrent model. During the whole training stage, the checkpoints trained by the unseen binary embeddings are not accessible.

**Evaluation details.** We input the unseen binary embeddings to the trained RPG to generate the parameters. The results of the original and generated unseen models are reported for comparison.

Table 10: Result comparisons of binary embedding change. Our approach can be aware of such change accurately. More results are shown in Appendix B.5.

| binary embedding (from seen set) | 1 | 0 | 1 | 1 | 1 | 0 | 1 | 0 | 0 | 0 |
|---|---|---|---|---|---|---|---|---|---|---|
| accuracy (%) | 98.0 | 99.1 | 98.5 | 94.4 | 98.1 | 92.2 | 99.2 | 97.0 | 97.1 | 99.1 |
| flipped embedding (from unseen set) | 0 | 1 | 0 | 0 | 0 | 1 | 0 | 1 | 1 | 1 |
| accuracy (%) | 92.3 | 98.9 | 94.0 | 85.4 | 92.2 | 90.8 | 99.3 | 95.3 | 97.6 | 98.1 |

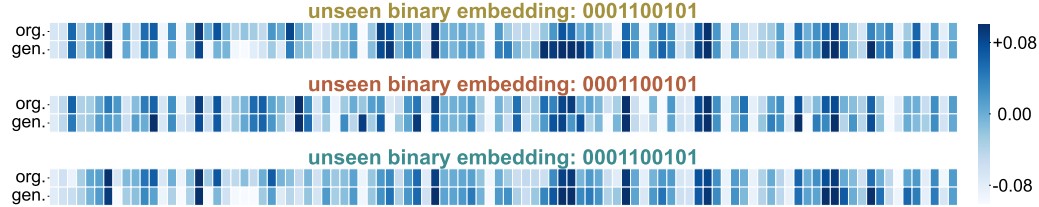

(a) Original and generated models with identical seen binary embeddings are compared. The three original models exhibit homogeneity, while the generated models display diversity.

(b) Original and generated models with 3 unseen binary embeddings are compared. The results confirm that our approach can learn high-performing parameter patterns even when they are not included in the training set.

Figure 6: Illustration of the parameters of original and generated models in seen and unseen embeddings. We select 100 parameters of the classification head and visualize its normalized values.

## 4.2 RESULTS OF GENERATING UNSEEN MODELS

**Performance comparisons.** We compare the results of our approach and original models on unseen binary embeddings in Tab. 9. Considering the space limitation, we randomly select 10 unseen binary embeddings for comparison. Notably, our approach yields commendable performance in these unseen tasks, even without being trained on the specific unseen embeddings. That demonstrates the strong practicality and potential of our approach in generating models under unseen tasks. The results of the remaining unseen binary embeddings and more analysis are shown in Appendix B.5.

**Perception of embedding changes.** In addition to comparing results, we further investigate our approach's ability to perceive embedding changes. We select two tasks with opposite binary embeddings in each element and report the results in Tab. 10. Our approach demonstrates a remarkable capacity to accurately detect changes in the tasks and generate corresponding model parameters. It is worth noting that the accuracy would hover around 50% if our approach were not aware of the embedding changes.

**Visualizations of original and generated model parameters.** To better understand the generated parameters, we visualize the original and generated models for both seen and unseen tasks in Fig. 6. For seen tasks, our approach generates diverse models compared to the original ones. Surprisingly, as shown in Fig. 6(b), we find that our approach can learn unseen parameter patterns. This demonstrates the potential generalization ability of our method.

## 5 RELATED WORKS

**Parameter generation.** The core idea of parameter generation is to learn the distribution of trained parameters. Stochastic neural networks (Sompolinsky et al., 1988; Bottou et al., 1991; Wong, 1991;

Schmidt et al., 1992; Murata et al., 1994; Graves, 2011) and Bayesian neural networks (Neal, 2012; Kingma & Welling, 2013; Rezende et al., 2014; Kingma et al., 2015; Blundell et al., 2015; Gal & Ghahramani, 2016) model the priors or probability distributions over the parameters. These approaches mainly employ the learned prior knowledge of parameters to improve robustness and generalization, and to mitigate overfitting and uncertainties in neural networks. However, these methods are limited by their poor generalization to large-scale or more complex real-world scenarios.

HyperNetworks (Ha et al., 2017), *i.e.*, a small network, is proposed to generate various architectures' parameters for a larger network. One year later, Smash (Brock et al., 2018) extends the range of the architectures via a memory read-writes scheme. With the development of diffusion technology, many works (Peebles et al., 2022; Chou et al., 2023; Erkoç et al., 2023; Wang et al., 2024; Soro et al., 2024; Lin et al., 2024; Li et al., 2024b; Jin et al., 2024) use the diffusion process to generate the neural network parameters. G.pt (Peebles et al., 2022) collects 23 million checkpoints as the training data and uses conditional diffusion to learn the distribution of the parameters. Besides the heavy cost of collecting so many checkpoints, G.pt can only generate less than 10K parameters. P-diff proposes unconditional diffusion to mimic the parameter updating and extend the size of generated parameters to 150K. COND P-DIFF (Jin et al., 2024) and Tina (Li et al., 2024b) introduce the task- or text-controlled parameter generation method. Unfortunately, the above methods have a common drawback: can not generate large-scale parameters, such as a whole ResNet, ViT, ConvNeXt, or LoRA. Therefore, our approach indeed largely increases the practicality of this field.

**Recurrent models.** Recurrent neural networks (RNNs) were first proposed to process sequential data, for example, the text. To tackle the vanishing gradient problem in early RNNs, long short-term memory (LSTM) (Hochreiter, 1991; 1997). In recent years, transformer-based models (Vaswani, 2017) starts to dominate the sequential data processing, due to their parallelized training and scalability. Although most transformers (Radford, 2018; Touvron et al., 2023) are used in the auto-regressive manner, they can be seamlessly converted into a recurrent model. However, the transformer-based models suffer have also been suffering from the the quadratic complexity problem. In recent year, various attempts, such as linear attentions (Wang et al., 2020; Choromanski et al., 2020), RWKV (Peng et al., 2023), Mamba (Gu & Dao, 2023; Dao & Gu, 2024), and xL-STM (Beck et al., 2024), have been don to tackle this problem. Different models above, which mainly focuses on modeling the language data, in this paper, we employ the recurrent model to build the relationship between parameters in neural networks.

**Diffusion models.** Diffusion models (Ho et al., 2020; Nichol & Dhariwal, 2021; Dhariwal & Nichol, 2021) witness an emergence in recent several years, due to their superiority in image generation. Many following works focus on improving the generation quality and efficiency of the diffusion model. For the first problem, Rombach et al. (2022) propose to conduct diffusion in the latent space, enabling high-resolution image synthesis. Peebles & Xie (2023) leverage the transformer (Vaswani, 2017) to explore scalability of diffusion models, proving the possibility of generating higher quality images with increasing size models. To solve the second problem, efficient samplers (Song et al., 2020; Lu et al., 2022; Song et al., 2023), efficiency models (Fang et al., 2023; So et al., 2024; Yang et al., 2023), and global acceleration approaches (Ma et al., 2024; Pan et al., 2024) are proposed. These method s facilitate generating high quality images with less computational and/or memory cost. Although significant progress on diffusion models has been made in image generation, how to improve quality and efficiency in large-scale parameter generation is still under-explored. In this paper, we propose the recurrent parameter generation model to tackle this problem.

## 6 DISCUSSION AND CONCLUSION

Our approach demonstrates promising results in large-scale parameter generation across various vision and language tasks. However, we acknowledge that achieving true 'AI creating AI' remains a distant goal. Firstly, while our method shows potential in generating models for unseen tasks, it currently faces limitations in generating parameters for novel model architectures. Secondly, our approach is constrained by modeling parameter relationships within a single task, potentially limiting its practical applicability. More importantly, future work should focus on simultaneously modeling parameter relationships across diverse architectures and tasks. Such an approach could yield a more powerful and versatile parameter generator, potentially advancing us closer to the 'AI creating AI' era. We hope our approach will inspire future research in this field.

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

We organize our appendix as follows.

**Experimental Settings:**

**Additional Analysis and Experimental Results:**

| training setting | configuration |
|---|---|
| **RPG-Tiny, RPG-Small, RPG-Base; Params. count < 50 million** | |
| the number of original models | 50 |
| batch size | 16 |
| optimizer | AdamW |
| learning rate | 3e-5 |
| training steps | 80,000 |
| weight decay | 1e-5 |
| mixed precision | bfloat16 |
| diffusion batch size | 1024 |
| **RPG-Large; Params. count > 50 million** | |
| the number of original models | 50 |
| batch size | 8 |
| optimizer | AdamW8bit |
| learning rate | 1e-5 |
| training steps | 120,000 |
| weight decay | 1e-5 |
| mixed precision | bfloat16 |
| diffusion batch size | 512 |

Table 11: Training recipe in detail. The definitions of RPG-Tiny, RPG-Small, RPG-Base, and RPG-Large can be found in Section A.3.

# A  EXPERIMENTAL SETTINGS

## A.1  TRAINING RECIPE

In this section, we provide detailed training recipes and supplementary information. The number of parameters generated by our approach ranges from approximately 3K to 200M. The significant disparity necessitates different training settings. Generally, as the number of parameters increases, the learning process becomes more challenging, requiring higher training costs, particularly for generating parameters beyond 50 million. Therefore, our training settings are divided into two categories: the default setting and the setting for parameters exceeding 50 million, as is shown in Tab. 11.

**Data parallelism:** When the number of parameters is less than 50 million, we adopt a single GPU to run the training process. For larger number of parameters, we employ distributed data parallelism to facilitate the training.

**Diffusion batch size:** In our approach, the diffusion model is shared across all tokens. Typically, all tokens can be fed as a single batch into the diffusion model for training. However, in practice, we randomly select a subset of tokens from a long sequence for training, rather than feeding all parts at once. This approach significantly reduces memory usage without compromising performance. The "diffusion batch size" in Tab. 11 refers to the number of tokens fed into the diffusion model during a single training iteration.

## A.2  DATASETS

In this section, we introduce the datasets used in the paper, including those for classification, semantic segmentation, object detection&instance segmentation, and commonsense reasoning.

**Classification**

- **ImageNet-1k** (Deng et al., 2009) is a large-scale visual database designed for use in visual object recognition research. It contains over 1 million images across 1000 categories and is widely used for training and benchmarking deep learning models.
- **CIFAR-10** (Krizhevsky & Hinton, 2009) dataset consists of 60,000 $32\times32$ color images in 10 different classes. It is commonly used for training machine learning and computer vision algorithms, providing a standard benchmark for image classification task.

**Semantic segmentation**

- **ADE20K** (Zhou et al., 2017) is a dataset for semantic segmentation and scene parsing, containing over 20,000 images annotated with pixel-level labels for 150 object categories. It is used to train models to understand and segment various objects and scenes in an image, making it valuable for applications in autonomous driving, robotics, and image editing.

**Instance segmentation & Object detection**

- **COCO** (Lin et al., 2014) dataset is a large-scale object detection, segmentation, and captioning dataset. It contains over 330,000 images, with more than 200,000 labeled instances across 80 object categories. COCO is widely used for training and evaluating models in object detection, segmentation, and image captioning tasks.

**Commonsense reasoning:**

- **BoolQ** (Clark et al., 2019): Yes/no questions based on natural passages.
- **PIQA** (Bisk et al., 2020): Questions about physical tasks and actions.
- **SIQA** (Sap et al., 2019): Questions about social interactions and implications.
- **HellaSwag** (Zellers et al., 2019): Choosing the correct ending for narratives.
- **ARC** (Clark et al., 2018): Multiple-choice science questions for grades 3-9.
- **OBQA** (Mihaylov et al., 2018): Questions requires multi-step reasoning, commonsense knowledge, and rich text comprehension.

### A.3 THE DETAILED STRUCTURE OF RECURRENT DIFFUSION

In this section, we provide specific details about the proposed recurrent model and diffusion model in RPG. More detailed configurations can be found in Tab. 12.

**Details of recurrent model.** By default, the recurrent model consists of two Mamba layers (Gu & Dao, 2023). As the increasing of parameters to generate, we need a larger recurrent model to capture the information in these parameters. The size of the recurrent model is mainly determined by the token size, which varies according to the number of parameters to be generated. Based on the token size, we categorize our model into four versions: Tiny, Small, Base, and Large.

**Details of diffusion model.** Following p-diff (Wang et al., 2024), our diffusion model adopts a one-dimensional convolutional architecture. The parameters of the diffusion model are significantly fewer than those of the recurrent model. We feed the prototypes from the recurrent model as conditions into the diffusion model by directly adding them to the feature map.

## B ADDITIONAL EXPERIMENTAL RESULTS AND FINDINGS

### B.1 DETAILED DISCUSSION WITH MORE RELATED WORKS

**Discussion with HyperRepresentation methods.** We mainly compare with three HyperRepresentation methods (Schürholt et al., 2022a; 2024; 2022b).

These methods use an autoencoder to learn the latent features of trained models, so they call the latent feature HyperRepresentation. This HyperRepresentation is then used for analyzing the model's performance or characteristics, or for sampling to generate new models or pre-trained parameters.

| module | setting | RPG-Tiny | RPG-Small | RPG-Base | RPG-Large |
|---|---|---|---|---|---|
| adequate number of parameters | | <50K | 50K~10M | 5M~50M | >50M |
| recurrent (Mamba) | d_model of 1st layer | 256 | 4096 | 8192 | 12288 |
| | d_model of 2nd layer | 256 | 4096 | 8192 | 16384 |
| | d_state | 32 | 128 | 128 | 128 |
| | d_conv | 4 | 4 | 4 | 4 |
| | expand | 2 | 2 | 2 | 2 |
| | parameter counts | 1.3M | 256M | 1018M | 3076M |
| diffusion (1D CNN) | encoder channels | (1, 32, 64, 128) | (1, 32, 64, 128) | (1, 32, 64, 128) | (1, 64, 96) |
| | decoder channels | (128, 64, 32, 1) | (128, 64, 32, 1) | (128, 64, 32, 1) | (96, 64, 1) |
| | token size | 256 | 4096 | 8192 | 16384 |
| | kernel size | 7 | 7 | 7 | 7 |
| | default solver | DDPM | DDPM | DDPM | DDIM |
| | sampling steps | 1000 | 1000 | 1000 | 60 |
| | $\beta$-start & $\beta$-end | (0.0001, 0.02) | (0.0001, 0.02) | (0.0001, 0.02) | (0.0001, 0.02) |
| | betas schedule | linear | linear | linear | linear |
| | number time steps | 1000 | 1000 | 1000 | 1000 |
| | parameter counts | 0.3M | 17M | 69M | 273M |

Table 12: Detailed information about four different sizes of recurrent diffusion. The *adequate number of parameters* implies that our model is usually adequate to generate parameters in that scale, which is empirical results instead of an exact rule. It also necessitates considering other factors such as parameter sensitivity.

- Schürholt et al. (2022a) utilizes kernel density estimation (KDE) to sample model parameters on the learned HyperRepresentation space. They also emphasize the importance of layer-wise loss normalization in the learning process of HyperRepresentation. This work achieves parameter generation in small CNNs from Model Zoos (Schürholt et al., 2022) with 2864 parameters.

- Schürholt et al. (2022b) focuses on using HyperRepresentation to sample the pre-trained model parameters. They also evaluate the ability of transfer learning by using a trained parameter autoencoder to initialize an unseen dataset. This work can be regarded as a cheap parameter initialization method.

- Schürholt et al. (2024) utilizes a sequential autoencoder for neural embeddings (SANE) to divide the neural network weights into subsets. Then, an autoencoder processes these subsets in a sliding window. This work can generate the entire parameter of ResNet-18. However, the performance of generated ResNet-18 is poor and exists a large gap with the original trained ResNet-18. For example, the performance of generated ResNet-18 is 68.6%, while the original model can obtain 94.2% accuracy on CIFAR-10 (see in Tab. 8).

We summarize the main differences as follows:

1. HyperRepresentation methods are hard to achieve comparable results as their original models that are used for training, but our approach obtains comparable results.

2. Schürholt et al. (2022b) focuses on parameter initialization while our approach targets to learn the distribution of high-performing neural network parameters.

3. SANE is the latest method among these three HyperRepresentation methods. However, SANE uses a sliding window to model the relationship of a small part of trained parameters. Our approach uses a recurrent model among all parameters.

4. Our approach can synthesize many popular vision and language parameters, such as ConvNeXt-L and LoRA parameters of LLaMA-7B, which is much larger than previous works.

**Details and limitations of G.pt.** A primary limitation of G.pt (Peebles et al., 2022) is the training data collection cost. By default, they collect 23 million checkpoints to train the parameter generator. Besides, they only evaluate the effectiveness of G.pt on small architectures, such as a low-dimensional MLP layer or a Convolutional layer with limited channels. The maximum number of generated parameters does not exceed 10, 000.

**Details and limitations of p-diff.** P-diff (Wang et al., 2024) directly flattens all parameters into a single-dimensional vector, disregarding the inter-layer parameter relationships. Furthermore, p-diff faces challenges in scaling up to large-scale parameter generation.

### B.2 TRAINING MEMORY COST ANALYSIS

In this section, we analyze the GPU memory utilization during training. GPU memory consumption is usually highly correlated with two factors: i) the size of the generative model and ii) the size of generated parameters. We analyzed the impact of these two factors on the GPU memory utilization during the training of our approach.

**GPU memory v.s. token size** We visualize the GPU memory usage with different token sizes in Fig. 7. As the token size increases, the scale of the recurrent model significantly grows, leading to a notable increase in GPU memory consumption. This implies that, when the performance of the generated models is comparable, we prefer to use models with smaller token sizes.

**GPU memory v.s. parameter counts** We conduct experiments to show the relationship between GPU memory and generated parameter counts in Fig. 8. In previous methods, the relationship between GPU memory consumption and the number of parameters in the generated model was quadratic (Schürholt et al., 2022a) or directly proportional (Wang et al., 2024). This limits their practicality and application range. In contrast, our approach demonstrates remarkable efficiency: with equivalent GPU memory usage, it can generate models with 34 to 960 times more parameters compared to previous methods.

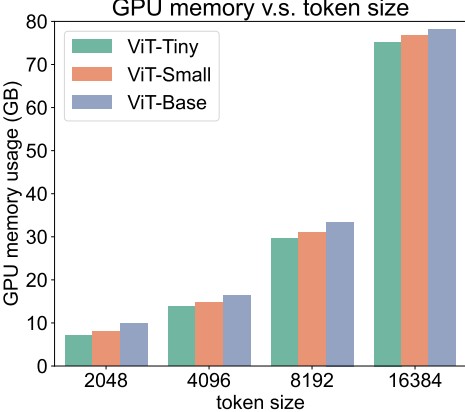
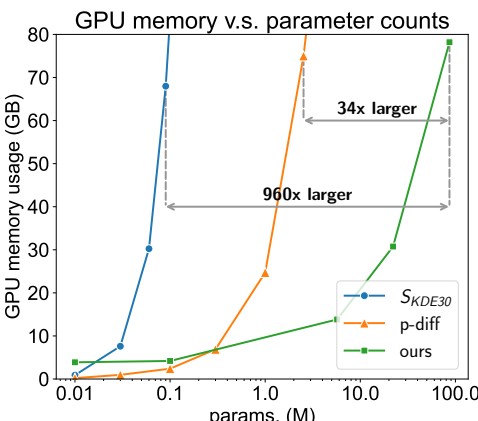

Figure 7: Visualization of GPU memory v.s. token size. GPU memory usage increases proportionally to the token size. Therefore, the token size cannot get larger infinitely; we need to choose an appropriate token size.

Figure 8: Visualization of GPU memory v.s. parameter counts. Our method can generate much more parameters than existing approaches e.g. $S_{KDE30}$ (Schürholt et al., 2022a) using a single NVIDIA H100 80G GPU.

### B.3 INFERENCE MEMORY COST AND SAMPLING TIME

In this section, we present more information about the sampling, including memory usage, inference time, and the balance between sequential and parallel inference.

In Tab. 7, we show the sampling time and memory usage for ViT-Base and ConvNeXt-L. Here, we present the sampling time and memory usage for other models. In Tab. 13, we adopt DDPM as the solver and conduct 1000-step sampling. Since the diffusion model in RPG is shared among all the

parameter tokens, we can adopt different inference modes to find a balance between memory usage and inference speed:

- **fully parallel:** All tokens are fed into the diffusion model simultaneously. This approach results in a high memory usage but achieves a high generation speed.

- **sequential:** Tokens are fed into the diffusion model one by one. This approach significantly reduces memory usage, as the model only occupies memory for inferring a single token at a time. This enable us to generate parameters of models listed on a GPU with less than 8GB of memory .

- **partially parallel (default):** In partial parallel mode, we set 256 tokens as a batch for the diffusion model inference. This approach significantly boosts speed with a slight increase in GPU memory usage, reaching an optimal trade-off between memory and speed. We adopt this as the default setting.

| metrics | inference mode | ResNet-18 | ResNet-50 | ViT-Tiny | ViT-Small | ConvNeXt-A |
|---|---|---|---|---|---|---|
| | sequential | 18.6 | 38.0 | 9.8 | 33.8 | 6.8 |
| time (minute) | partially parallel | 1.8 | 3.3 | 1.1 | 2.9 | 0.9 |
| | fully parallel | 1.7 | 3.3 | 1.1 | 2.9 | 0.9 |
| | sequential | 6.3 | 6.4 | 6.2 | 6.4 | 6.2 |
| memory cost (GB) | partially parallel | 10.3 | 10.5 | 10.3 | 10.5 | 10.3 |
| | fully parallel | 30.8 | 50.5 | 19.4 | 45.9 | 15.2 |

Table 13: We show the inference time and memory cost under different inference modes. All information in this table is collected from a single NVIDIA H100 80G GPU. We report the time and memory required to generate a single model.

Based on the results in Tab. 13, our approach can be flexibly applied to many other GPUs as it can achieve a good trade-off between memory and time.

### B.4    PARAMETER SPACE ANALYSIS

In this section, we demonstrate that our method offers significant advantages over simply adding noise to the original models. Following the p-diff (Wang et al., 2024), we choose Intersection of Union (IoU) as the metric for measuring similarity. It compares the agreement of output results from classification models across a large number of samples to evaluate the similarity. We calculate the IoU of one model with all the original models and select the maximum IoU value (nearest neighbor) as the measure of similarity.

In Fig. 9, we compare adding various levels of noise to the original models with the models generated by our method in terms of accuracy and similarity. As the noise level increases, the similarity of the models decreases, but the accuracy decreases as well. The points representing our generated models are distributed in the *upper left* region relative to the area with added noise, indicating that our models can enhance diversity while maintaining accuracy.

### B.5    MORE RESULTS OF SECTION 4

**Results of generating models for unseen tasks.** In Section 4, we show the potential of our approach in generating models for unseen tasks. In this part, we provide more results. First, we compare the performance of original and generated models using all unseen embeddings in Tab. 14. Results demonstrate that our approach consistently achieves good results in unseen tasks.

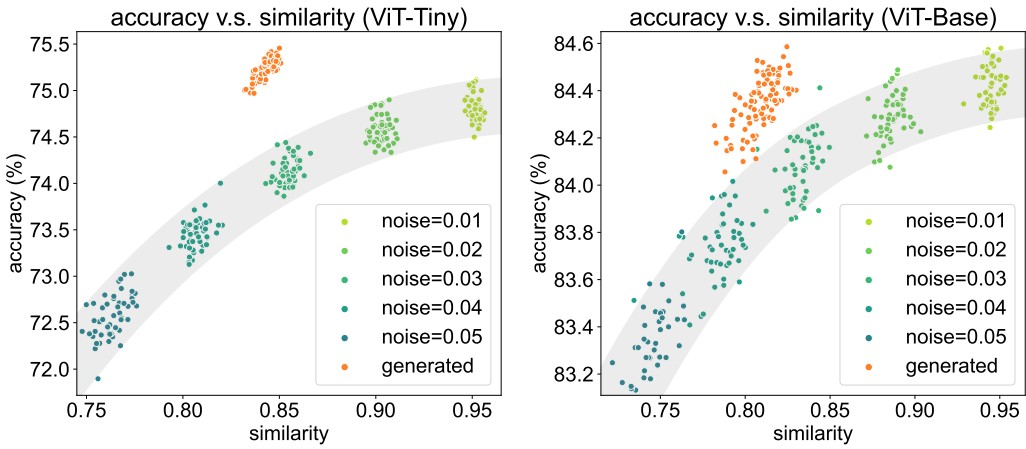

Figure 9: The figures show the balance between accuracy and similarity. The shaded area indicating the approximate range of noise added checkpoints. These two plots demonstrate that our method has a more favorable performance in terms of both accuracy and similarity compared to adding noise to the original checkpoints.

| unseen binary embeddings | original | best acc. | average acc. | standard deviation |
|---|---|---|---|---|
| 0 1 0 0 0 1 0 1 1 1 | 97.3 | 94.4 | 93.9 | 0.6 |
| 0 1 1 1 1 1 0 1 1 0 | 98.1 | 96.6 | 94.9 | 2.1 |
| 0 0 1 1 1 0 1 1 1 0 | 97.4 | 95.0 | 94.2 | 1.1 |
| 0 1 0 1 1 1 1 1 1 1 | 98.4 | 96.1 | 95.8 | 0.3 |
| 0 0 1 0 0 0 0 0 0 0 | 98.9 | 96.6 | 95.2 | 2.3 |
| 0 0 0 1 1 0 0 1 0 1 | 96.7 | 92.9 | 91.6 | 1.1 |
| 1 1 1 1 1 0 1 0 0 1 | 97.6 | 94.8 | 94.1 | 0.7 |
| 1 0 1 0 0 0 0 0 1 1 | 98.1 | 95.7 | 91.8 | 3.7 |
| 0 1 0 0 0 1 0 1 1 0 | 97.1 | 93.6 | 90.7 | 4.3 |
| 1 1 0 0 0 1 1 0 0 1 | 97.0 | 94.0 | 90.1 | 3.6 |
| 1 0 1 0 0 0 1 1 0 1 | 97.3 | 91.3 | 90.7 | 0.8 |
| 0 1 1 1 1 0 0 0 1 0 | 96.3 | 95.4 | 89.4 | 6.3 |
| 1 1 0 1 1 1 0 1 0 0 | 97.6 | 92.6 | 90.5 | 3.2 |
| 0 1 1 1 0 0 1 1 1 0 | 96.3 | 90.8 | 89.1 | 1.9 |
| 0 1 0 0 1 1 1 0 1 0 | 96.3 | 91.9 | 88.4 | 4.4 |
| 0 0 1 0 0 0 1 1 0 1 | 97.5 | 93.7 | 88.0 | 5.6 |
| 0 0 0 1 1 0 1 1 1 1 | 96.5 | 90.8 | 85.5 | 7.0 |
| 1 0 0 1 0 0 1 1 0 1 | 96.4 | 86.7 | 83.7 | 3.6 |
| 1 0 0 1 0 1 0 0 0 0 | 97.7 | 85.6 | 83.2 | 2.0 |
| 0 0 1 1 0 0 0 1 0 1 | 96.3 | 90.2 | 79.2 | 9.6 |

Table 14: Performance comparisons between original and generated models in unseen tasks. Our approach show its strong ability to generate models that could satisfy unseen tasks.

**PCA visualization of classification head parameters.** We also provide a visualization of the parameters of the classification head (a two-layer fully connected structure with total 38,976 parameters) for 1022 tasks as described in Section 4 using Principal Component Analysis (PCA), which presents the structure of the parameter space in Fig. 10(a). Our generated model achieves an average accuracy of 91.2% across all binary classification tasks, which indicates that our method has effectively learned this structure. Furthermore, we evaluate the parameters corresponding to unseen tasks and compared their positions in Fig. 10(b) between the original and generated parameters. It is

noteworthy that, even though the original parameters of these tasks are not included in the training data, the generated parameters consistently appeared in close proximity to the original ones. This observation further highlights the capability of our method to model the structure of the parameter space, even for tasks not previously encountered.

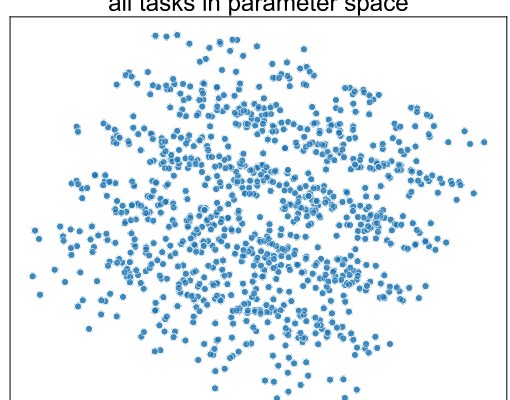

all tasks in parameter space

(a) Visualization of the classification head of all 1022 tasks. This reveals that there is an inherent structure among the high-dimensional parameters.

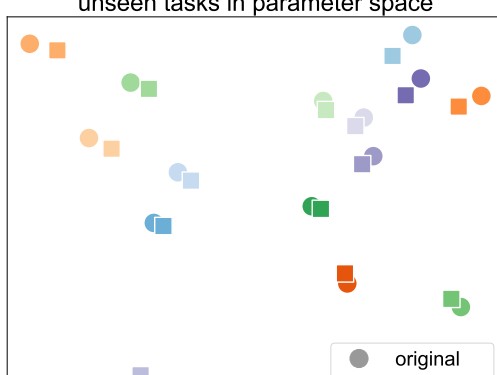

unseen tasks in parameter space

(b) Visualization of the classification head in some unseen tasks. Parameters associated with the same task are indicated by a consistent color.

Figure 10: Principal Component Analysis (PCA) visualization of the classification head. The figures demonstrate the presence of an inherent structure in the high-dimensional parameter space and highlight our method's effectiveness in capturing this structure for unseen tasks.

### B.6 THE EFFECT OF PERMUTATION STATE

We demonstrate the importance of the permutation state in mitigating model symmetry issues (Badrinarayanan et al., 2015; Kunin et al., 2021). Model symmetry refers to a characteristic in neural networks where parameters exhibit permutation symmetry—swapping certain parameters changes the parameters themselves, but the model output remains unchanged. This feature brings a significant challenge in modeling and understanding of neural network parameters. During parameter generation training, using checkpoints from different random seeds can introduce this problem.

To demonstrate the effectiveness of the permutation state in RPG, we conduct experiments by training our RPG with and without it, using checkpoints (ViT-Tiny on CIFAR-10) from various random seeds. Results in Tab. 15 demonstrates that our permutation state design obviously alleviates the learning difficulties caused by parameter symmetry. It enables the generated parameters to achieve performance comparable to the original checkpoints, even with 10 random seeds. However, our design may not be the optimal solution and the performance declines when the seeds increase to 20. This encourages us to further explore this issue in the future.

| method | original | w/ permutation state | | | | w/o permutation state | | | |
|---|---|---|---|---|---|---|---|---|---|
| No. of seeds | − | 1 | 3 | 10 | 20 | 1 | 3 | 10 | 20 |
| accuracy (%) | 88.1 | 88.1 | 88.1 | 88.2 | 29.1 | 88.0 | fail | fail | fail |

Table 15: The number of random seeds used when preparing checkpoints is denoted as the *No. of seeds*. We compare the differences between using the permutation state (*w/ permutation state*) and not using the permutation state (*w/o permutation state*). The results in this table are obtained from ViT-Tiny on CIFAR-10 dataset.

## B.7 PARAMETER SENSITIVITY V.S. PERFORMANCE

According to conventional understanding, larger parameter quantities are generally more challenging to learn. However, our experiments reveal that this rule is not absolute and demonstrates instability in learning some small model parameters.

This motivates us to investigate the relationship between parameter sensitivity and generation quality. Specifically, we add Gaussian noise with weights of 0.01, 0.10, and 1.00 to the original parameters to measure model sensitivity, as shown in Tab. 16. We observe that as noise weight increases, performance decreases for all models, with smaller models being more affected than larger ones. This indicates that smaller models are relatively more sensitive. Additionally, we notice that the performance gap between the original and generated models widens as model size decreases. This demonstrates a strong correlation between a model's sensitivity and the difficulty of generating its parameters.

| model | params. (M) | sensitivity | accuracy decline | | | |
| --- | --- | --- | --- | --- | --- | --- |
| | | | ours | noise (0.01) | noise (0.10) | noise (1.00) |
| ConvNeXt-A | 3.7 | +++ | 0.85 | 62.83 | 0.60 | 0.03 |
| ResNet-18 | 11.7 | ++ | 0.39 | 53.56 | 0.46 | 0.00 |
| ViT-Base | 86.6 | + | 0.09 | 5.39 | 0.02 | 0.00 |

Table 16: The accuracy decline reflects the accuracy gap between the original model and the generated model or the model after adding noise. We add Gaussian noise with weights of 0.01, 0.10, and 1.00 to the parameters to measure model sensitivity. Results demonstrate that smaller models are relatively more sensitive than larger ones. The more plus signs (+) , the higher the sensitivity.

## B.8 TRANSFORMER (IN TAB. 5) EXPLANATION

It is noteworthy that we use a transformer encoder with causal attention, rather than the commonly used auto-regressive transformer decoder structure in large language models, because our model generates tokens recurrently rather than auto-regressively.