# OpenReview forum: "Recurrent Diffusion for Large-Scale Parameter Generation"
_ICLR.cc/2025/Conference — ICLR 2025 Conference Withdrawn Submission_

### Official Review · Reviewer_oFtj · 2024-10-27

**Soundness:** 1
**Presentation:** 3
**Contribution:** 2
**Rating:** 3
**Confidence:** 4

**Summary:**

The authors present a method to generate larger scale (~100M parameters) neural network weights.
They do so by splitting the neural networks weights to groups, each defined as a token. They then use these tokens to condition a diffusion process to generate neural network weights .
The network weights generated with this method achieve competitive performance with normally-trained models.

**Strengths:**

1. The problem tackled in the paper is important - specifically, an attempt to scale up generation of neural network weights.
2. The approach taken by the authors to "split up" parameter generation to parts by treating the entire generation task as an autoregressive task looks novel.
3. The paper is easy to follow and -- in a high level -- is written well (there are many tweaks and proofing required. These can improve the writing, see relevant point in "Weaknesses" tab).

**Weaknesses:**

The main weakness of the paper is lacking details.

Several examples of this in early sections are -
	1. Which weights are generated exactly? Is there an assumption that all generated weights w_i are of the same type (i.e. same shape, used for the same type of layer (conv, attention, etc))?
	2. While the method allows for lower memory consumption in parameter generation, how does it compare to existing methods in *inference* runtime? This detail is missing, and is important for comparison between this method and existing methods.
	3. The motivation behind some decisions is unexplained: for example, why do the authors choose to normalize weights w_i? Is it grounded in previous work / did the authors find this improves generation? A further explanation would be nice.
	4. In section 2.3, lines 142-146, the motivation behind the permutation states is a bit unclear. Perhaps a more detailed explanation (2-3 sentences) on network symmetries is in order.
	5. In equation 3, it is unclear why the arrow starts at K[i], since the values of the positional embeddings e[1]…e[i] do not depend on it (unlike in equation 2, where it is clear that the different k[i] are parts of w[i]).

There is also a lack of (crucial) details in the experiments section -
	1. For example, the authors claim that relations between different "parameter groups" are important and that mixing up weight groups between models degrades performance (fig 2). However, the authors don't explain well what are the parameter groups they mix.
	2. The meaning of "fail" in the ablation results (Table 4) is unclear.
	3. The paragraph on "Results on commonsense reasoning" (lines 257-262) is lacking. More details regarding the finetuning process are required. For example, how many training checkpoints are used. Also, unless otherwise mentioned, it seems the generated weights are evaluated on the same data that the DoRA-trained-checkpoints were trained on.
	4. In Table 8, some comparisons to other weight-generation methods are missing (e.g. The accuracy of SANE on CNN(s)). This makes the obtained results seem less trustworthy.

Additionally, many sections require proofing. Several examples -
	1. Lines 188-189, remove the "the" before task names.
	2. In the related work section - It is better to use the same tense across the paragraph. There are also several writing errors (e.g. line 501 "or"->"of"; line 505 "the text" -> "text").
	3. Line 370 -> "previous works are hard to achieve comparable performance"
	4. "I", the maximal layer index, should be explicitly defined when it is first used (also applies to other notations that aren't properly defined).
	5. In the captions of tables 1 and 4, it is written "Bold entries are best results" but no results are marked.
	6. Table 5 "traning" -> "training".

**Questions:**

1. For a future revision, I believe It would be nice to see an analysis on different tokenization techniques. Since this is the main novelty of the approach, it would be cool to compare the authors' approach of tokenization within the layer to other methods, which might tokenize across layers somehow (given the previously-shown repetition of similar weight vectors across layers).

2. In Table 1, what is "medium" accuracy? Generally, I think it would be better to convert this table to a plot showing the mean and std accuracy only. No need for max / min etc.

---

### Official Review · Reviewer_EuFH · 2024-11-03

**Soundness:** 4
**Presentation:** 3
**Contribution:** 3
**Rating:** 8
**Confidence:** 4

**Summary:**

This paper introduces the Recurrent Diffusion for Large-Scale Parameter Generation (RPG) method, which divides the neural network to be generated into different parts. It uses a recurrent neural network to learn the relationships between different parts and employs a diffusion network to generate parameters for each part separately. This partitioning approach addresses the OOM issue of generating parameters for large-scale models. The proposed method has been validated on models and tasks of various scales. Detailed ablation studies confirm the effectiveness of key designs such as permutation state, position embedding, tokenization, and the recurrent model. The method outperforms existing approaches across different model sizes. Additionally, the paper demonstrates that the generated network parameters perform well on unseen tasks, further highlighting the significance of parameter generation.

**Strengths:**

- Innovatively proposes using an RNN to model the relationships between different parts and then generating parameters for each part separately, solving the out-of-memory (OOM) problem. The method is simple and easy to understand. Its effectiveness is thoroughly validated through detailed ablation studies and analytical experiments.
- Cleverly designs unseen tasks to test the characteristics of the generated model parameters, emphasizing the importance of parameter generation.
- Provides detailed training settings and computational resource usage.
- A particularly appealing aspect to the reviewer is Section B.6, where permutation states are added to models with different random seeds to successfully handle parameter symmetry. This clearly suggests that the method does not simply memorize model parameters. Interestingly, models trained with 10 seeds + permutation states show slightly higher generation accuracy, suggesting that scaling the number of better-performing models can lead to even better parameter generation.

**Weaknesses:**

- The introduction of inference details is unclear. Does repeating the experiment ten times involve changing the permutation state used, or just altering some random state?
- The author mentions in the limitations that the method is still limited to generating parameters for models with the same architecture and task.

**Questions:**

- Regarding the permutation state:
    1. How is the permutation state used during the inference phase, as mentioned in the weaknesses section?
    2. How much do the effects differ among different permutation states? Is there an issue if the permutation states used during training differ significantly?
- Besides handling unseen tasks, the reviewer is curious whether the generated parameters offer better performance in out-of-distribution (OOD) tasks or robustness compared to the original model?

---

### Official Review · Reviewer_g28v · 2024-11-04

**Soundness:** 3
**Presentation:** 2
**Contribution:** 2
**Rating:** 5
**Confidence:** 3

**Summary:**

The paper introduces a novel approach called Recurrent Diffusion for large-scale Parameter Generation (RPG), aiming to efficiently generate large-scale neural network parameters. The authors address the challenge of scaling parameter generation by dividing trained network parameters into non-overlapping tokens and employing a recurrent model to learn the relationships among these tokens. The outputs of the recurrent model serve as conditions for a diffusion model, which generates the neural network parameters. This hierarchical method allows for the generation of large-scale models, such as ConvNeXt-L and LoRA parameters of LLaMA-7B, using only a single GPU. Experimental results demonstrate that the generated parameters perform comparably to the original trained models across various architectures and tasks, including classification, semantic segmentation, object detection, and language tasks. Additionally, the paper explores the potential of RPG to generate models for unseen tasks on CIFAR10.

**Strengths:**

- The paper presents a new approach for parameter generation combining autoregression and diffusion. They use SSMs (Mamba) to easily and effectively perform large-scale parameter generation.
- The paper also presents a method for parameter tokenization, which they show later on performing significantly better than tokenization methods used by previous works.
- The paper includes interesting ablation studies that show the contributions of different presented components.

**Weaknesses:**

- The approach is very limited in novelty. Autoregressive models feeding embeddings into a diffusion model is not new in general.
- The paper lacks a thorough analysis of whether the method genuinely learns to generalize the parameter distribution or simply memorizes the training data. There is a need for more evidence to show that the generated parameters are not merely reproducing the training checkpoints.
- The evaluation on unseen tasks using CIFAR-10 with binary embeddings is not clearly explained and may not convincingly demonstrate the method's generalization capabilities. The experimental setup seems artificial and may not reflect practical or meaningful scenarios.
- Certain values from previous works (eg. Table 8, p-diff, ViT-Base) are presented as OOM. This is despite the previous work (p-diff) successfully generating ViT-Base parameters on a 40GB A100 (according to their paper[1]). More explanation here would be appreciated.
- The paper's presentation could be significantly improved. Some sections lack clarity, and important details are either missing or not well-explained. This makes it difficult to fully understand the methodology and reproduce the experiments. Better organization and clearer explanations are necessary.

**Questions:**

- Can you provide a more detailed analysis on how the generated parameteres differ from original models? For instance, any measure of diversity of the generated parameters to demonstrate that the model might be doing something more than basic memorization?
- Please clarify the experimental setup on CIFAR-10. How does assigning random binary labels to CIFAR-10 categories and generating models for these tasks meaningfully demonstrate generalization?
- Regarding Table 8, can you explain why methods like p-diff are reported as OOM for ViT-Base when their original papers claim successful generation of such models? Have you conducted these experiments yourself, and under what settings? Clarifying this would ensure a fair comparison.

---

### Official Review · Reviewer_qHcP · 2024-11-04

**Soundness:** 2
**Presentation:** 3
**Contribution:** 2
**Rating:** 3
**Confidence:** 3

**Summary:**

The paper presents a novel method for large-scale parameter generation, termed Recurrent diffusion for large-scale Parameter generation (RPG). Distinct from previous approaches, RPG incorporates parameter correlations and employs a recurrent model to learn the interrelationships among non-overlapping parameter tokens. The recurrent network receives tokenized parameters, generated from layer divisions and normalized, along with positional encodings that indicate the layer index and token position within the layer. To produce the parameters, the output of the recurrent model, referred to as the 'prototype,' is subsequently input into a diffusion model. This method is evaluated across various tasks and architectures, in both same-task and task-transfer settings.

**Strengths:**

- The paper introduces a new approach that combines recurrent neural mechanisms with diffusion-based generative modeling to effectively capture and stabilize dependencies between model parameters. This approach specifically targets complex parameter interdependencies that arise in large models, and experimental results suggest it leads to stable and consistent parameter generation, enhancing model robustness.
- The method is validated across diverse architectures, including but not limited to ResNet, ViT, and LLaMa-7B, demonstrating its applicability to both vision and language tasks. The generated parameters consistently match or closely approximate the performance of the original models, with results achieved on a single GPU. This supports the approach's generality and computational efficiency, making it viable for a range of use cases in both academic and applied contexts.
- The paper shows a creative reimplementation of classification task to show that one can treat parameter generation as a conditional generative task, which shows promising results on par with original performance on unseen tasks in Section 4

**Weaknesses:**

- **Practical Limitations**: The method exhibits significant practical constraints for both similar and generalizable tasks. The reliance on numerous checkpoints (50 checkpoints) from fully trained models raises questions about its application to novel architectures. Although preliminary exploration of task transfer is presented, the necessity of training many models for seen tasks and the requirement for clearly defined task embeddings to relate seen and unseen tasks limits practical applicability.

- **Task Embeddings**: Section 4 relies on predefined task embeddings, which may not adequately capture the complexity of real-world tasks. The experimental setup is limited to binary classification on CIFAR-10, restricting task diversity. Additionally, the rationale behind the choice of three checkpoints and the methodology for dividing seen and unseen embeddings is unclear. Reporting results for only ten unseen embeddings appears insufficient for robust validation.

- **Unsupported Claims**: While standard deviations are reported in Table 1, some ablation studies lack this detail, particularly Table 4b, where the claim that learnable embeddings outperform others is only weakly supported by a minimal score difference of 0.1 across all models.

**Questions:**

1. How are training parameters determined? What is the minimum requirement for model training or task generalization training? Can this approach function effectively with fewer checkpoints? Furthermore, how does the total computational cost compare to traditional training methods, especially considering the need for 50 checkpoints?

2. Can RPG be extended to generate parameters for novel architectures?

3. How robust is RPG to changes in task complexity? Are there any multi-class classification tasks that utilize different types of embeddings?

4. From a conceptual standpoint, what is the added value of generating parameters if checkpoints already exist? What necessary steps are needed to make parameter generation a practical solution?

---

### Official Review · Reviewer_3uT4 · 2024-11-04

**Soundness:** 3
**Presentation:** 2
**Contribution:** 3
**Rating:** 5
**Confidence:** 2

**Summary:**

The paper proposes an approach for large scale neural network parameter generation. The idea is that given a model architecture and some checkpoints from training/tuning a model, one can have a more lightweight model predict the weights/parameters of an optimal model.
The specific idea proposed in the paper is to use a recurrent neural network and a (1D) diffusion process together to predict parameters. This is done by first tokenizing the parameters in each layer and padding them appropriately to be passed into a recurrent model (along with position encodings that inform the layer index and the position within that layer). The output from the recurrent model then goes through a diffusion process to predict tokens for the final parameters, this way the diffusion is conditioned on the input tokenized parameters.

The paper then evaluates the approach on several vision and language tasks image classification, object detection, segmentation, and common-sense reasoning. They evaluate several existing model architectures for each of these tasks, and predict the parameters for these models (CNNs, ViTs, LoRa/DoRa params for LLaMA).

**Strengths:**

* The paper introduces a technique for parameter generation that appears to scale to larger models than some previous works.
* The evaluation considers models and model architectures for different vision tasks and a language task.

**Weaknesses:**

W1. While figure-2 provides a reasonable motivation, it is not clear from the main paper exactly what are the trade-offs that one should consider for parameter generation. E.g. Why should one consider parameter generation as opposed to training/tuning a model on a given dataset? Is there any advantage in terms of compute costs, if so which stages of the proposed RPG method contribute to it?

W2. Some related works necessary to understand the main contributions of the paper are in the appendix. It would be good to specifically highlight works most similar to your work and how exactly your work is different, especially in the context of Q1.

W3. Section 4 on evaluation of unseen tasks is hard to understand. (see more specific questions under “Questions”). The CIFAR-10 dataset, as I understand, has each image corresponding to 1 category, so it is not at all clear when you suggest 2^10 potential classes on the dataset. Also, by unseen task, it appears that you are still doing classification and not some new kind of task that these models have not done before.

W4. Sec 3.2 Results is missing comparison with existing methods on similar tasks. How well do other methods do on some of these tasks? (Table 8 has 1 set of comparisons but that seems somewhat restricted). Can you share more on why existing methods are not compared? Is it because they cannot be used on all these tasks/models? Why/why not?

**Questions:**

Q1. Is it correct to say that in your approach, given the model architecture you generate parameters? Or is the architecture also generated as part of the tokenization and recurrent generation?

Q2. Can describe briefly the motivation for what are the advantages of parameter generation as opposed to training or tuning a model from scratch? It appears that your model still needs several checkpoints from trained models so that it can learn parameters. So it’s not clear at what stage one would do parameter generation. Also specifically for RPG, how does this differ compared to existing methods?

Q3. Can you clearly state the trade-off in terms of compute and effort in doing RPG vs existing parameter generation methods. You do mention your proposed technique allows predicting parameters for larger models, but not specifically how much effort it is to do RPG vs just regular tuning and hyper param optimization. Also, why is there no comparison to hyper parameter optimization strategies and techniques (e.g. AutoML?)


Q4. The CIFAR-10 dataset as I understand has each image corresponding to 1 category, so what exactly do you mean when you do binarization of tasks? Can you be specific about what you mean by a task here? In some places you use unseen models (line. 431) but other places task, that adds to the confusion.

Suggestions

It is worth front loading related works to give greater context of the proposed idea and motivating it better. Similarly parts from appendix needs to be moved up to motivate your contributions better.
Lines 142-146 it's worth describing briefly what you mean by neutral network symmetries and explain how exactly they have an effect on parameter generation.

---

### Official Review · Reviewer_d3kn · 2024-11-05

**Soundness:** 3
**Presentation:** 3
**Contribution:** 3
**Rating:** 5
**Confidence:** 3

**Summary:**

This paper introduces a new way of generating model parameters for large models using a combination of a recurrent model (Mamba) and diffusion. The input to the recurrent model consists of two pieces of metadata: The position of the parameters within the network, and the permutation state. The output (and training target) of the model consists of "parameter tokens", i.e., small subsets of the network parameters. This approach allows the parameters to be generated (and the network to be trained) piece-wise using constant memory requirements (rather than scaling with the size of the target network). The paper also explores the ability of this approach to generate models for unseen tasks by having metadata of the target task as an additional input.

**Strengths:**

The paper is clearly written, the model straightforward and the experimental results quite convincing. The authors performed a series of insightful ablations (e.g., different sequence models, with/without sequence model, and different positional embeddings).

**Weaknesses:**

My understanding is that the SANE method (Schürholt et al., 2022), which is mentioned in the related works section, should have similar scaling properties as RPG. It seems to me that OOM errors can be avoided by changing their tokenization scheme and/or reducing the window size of the sequential autoencoder, so I wonder if the OOM errors in table 8 are a bit misleading. What hyperparameters were chosen exactly?

The main differences I see between RPG and SANE in the use of Mamba vs. a sequential autoencoder for learning the token relationships, the use of a diffusion model instead of a transformer model for mapping embeddings to output weights, and the use of position embeddings. I'd be curious to see these differences ablated, e.g., how does SANE perform when using RPG's tokenization scheme, or how does RPG perform when using transformers instead of a diffusion model. This would provide some insight into where the performance boost compared to SANE is coming from.

Because my main concern is that the proposed model seems pretty good--it's clearly very beneficial for memory usage to generate one parameter token at a time--but that it's a bit unclear to me what components of the model are actually essential in reaching the shown performance: Clearly the exact sequence model (Mamba vs. transformers) doesn't matter too much (table 5) and the tokenization scheme effects seem quite minor (figure 4), so what explains the big gap with SANE (for example, table 8)? Without this insight the paper provides a good model to use of the shelf, but doesn't really provide the scientific understanding of why the model works.

A second concern is that the experiment design for unseen tasks seems a bit artificial. Although it provides a proof of concept of the generating network being able to understand some form of task descriptions, the actual practical application of inputting binary vectors to describe which classes should be positive/negative seems limited. Perhaps a more practical experiment would have included an LLM that maps a task description to an embedding?

**Questions:**

See above.

---

### Note · Authors · 2024-11-14

**Comment:**

we will improve it

**Withdrawal Confirmation:**

I have read and agree with the venue's withdrawal policy on behalf of myself and my co-authors.